# Spinal Metastasis in a Patient with Supratentorial Glioblastoma with Primitive Neuronal Component: A Case Report with Clinical and Molecular Evaluation

**DOI:** 10.3390/diagnostics13020181

**Published:** 2023-01-04

**Authors:** Michal Hendrych, Peter Solar, Marketa Hermanova, Ondrej Slaby, Hana Valekova, Marek Vecera, Alena Kopkova, Zdenek Mackerle, Tomas Kazda, Petr Pospisil, Radek Lakomy, Jan Chrastina, Jiri Sana, Radim Jancalek

**Affiliations:** 1First Department of Pathology, St. Anne’s University Hospital Brno and Faculty of Medicine, Masaryk University, 602 00 Brno, Czech Republic; 2Department of Neurosurgery, St. Anne’s University Hospital Brno and Faculty of Medicine, Masaryk University, 602 00 Brno, Czech Republic; 3Department of Biology, Faculty of Medicine and Central European Institute of Technology (CEITEC), Masaryk University, 625 00 Brno, Czech Republic; 4Central European Institute of Technology (CEITEC), Masaryk University, 625 00 Brno, Czech Republic; 5Department of Radiation Oncology, Masaryk Memorial Cancer Institute Brno and Faculty of Medicine, Masaryk University, 656 53 Brno, Czech Republic; 6Department of Comprehensive Cancer Care, Masaryk Memorial Cancer Institute and Faculty of Medicine, Masaryk University, 656 53 Brno, Czech Republic

**Keywords:** glioblastoma, metastasis, NF1, NOTCH3, ARID1A, mutation

## Abstract

Glioblastoma (GBM) is regarded as an aggressive brain tumor that rarely develops extracranial metastases. Despite well-investigated molecular alterations in GBM, there is a limited understanding of these associated with the metastatic potential. We herein present a case report of a 43-year-old woman with frontal GBM with primitive neuronal component who underwent gross total resection followed by chemoradiation. Five months after surgery, the patient was diagnosed with an intraspinal GBM metastasis. Next-generation sequencing analysis of both the primary and metastatic GBM tissues was performed using the Illumina TruSight Tumor 170 assay. The number of single nucleotide variants observed in the metastatic sample was more than two times higher. Mutations in *TP53*, *PTEN*, and *RB1* found in the primary and metastatic tissue samples indicated the mesenchymal molecular GBM subtype. Among others, there were two inactivating mutations (Arg1026Ile, Trp1831Ter) detected in the *NF1* gene, two novel *NOTCH3* variants of unknown significance predicted to be damaging (Pro1505Thr, Cys1099Tyr), one novel *ARID1A* variant of unknown significance (Arg1046Ser), and one gene fusion of unknown significance, *EIF2B5-KIF5B*, in the metastatic sample. Based on the literature evidence, the alterations of *NF1*, *NOTCH3*, and *ARID1A* could explain, at least in part, the acquired invasiveness and metastatic potential in this particular GBM case.

## 1. Introduction

Glioblastoma (GBM) is the most common glial brain tumor in adults (median age of 65 years), which accounts for nearly 50% of malignant primary brain tumors [1] and represents a disease with a dismal prognosis. GBM is characterized by rapid infiltrative spread within the central nervous system. Following surgical resection, GBM progresses from residual tumor cells in approximately half of the patients even before the initiation of oncological treatment [2] and inevitably leads to death either due to brain stem infiltration or brain swelling [3]. Median survival corresponds to 14–16 months, notwithstanding the aggressiveness of oncological treatment [4,5]. Despite the aggressiveness and invasive nature of GBM, extracranial metastases are rare clinical events. The incidence of symptomatic GBM metastasis has been estimated in approximately 2% of patients by different studies [6,7]; however, systemic GBM metastases were detected in 6–27% of autopsy series [8,9]. 

The aim of our report is to present a case of spinal GBM metastasis, describe its molecular background, and search for differences in gene mutation profiles between the primary and metastatic disease to outline possible molecular pathways of metastatic spread. This topic is becoming ever more relevant since there seems to be an increasing trend of extracranial metastasis in GBM, likely due to the steadily increasing overall survival in GBM patients [4,6]. 

## 2. Case Report

A 43-year-old woman with organic psychosyndrome was examined by a neurologist and subsequently referred to a neurosurgeon with an MRI finding of an intra-axial contrast-enhancing lesion in the left frontal lobe with perilesional edema and midline shift (Figure 1A). The patient underwent MRI-navigated fluorescence-guided surgery using 5-aminolevulinic acid (5-ALA), which led to the opening of the left lateral ventricle due to tumor infiltration of the ventricular wall. Tissue samples from the tumor core and resection margins were collected separately for histological evaluation. Based on an MRI obtained within 48 h after surgery, the extent of resection was marked as a gross total resection (Figure 1B).

The sample from the tumor core displayed diffusely infiltrating hypercellular glioma with numerous foci of geographic and palisading necrosis as well as multiple microvascular proliferations. Tumor cells of astrocytic differentiation showed marked pleomorphism, including multinucleated giant cells and sparse regions with epithelioid morphology alternating with regions formed by small cells growing in solid formations or even areas formed by spindle-shaped cells (Figure 2). No foci of true rosettes or pseudorosettes were detected. Immunohistochemically (IHC), the primary tumor cells displayed expression of the GFAP, which varied among different components (Figure 3A,B), and proliferation index Ki-67 (MIB-1) diffusely in 40% of tumor cells, in hotspots reaching up to 99% (Figure 3C). Using the mutation-specific antibody, IHC did not show the R132H mutation in the *IDH1* gene. Subsequent Sanger sequencing of the *IDH1* (codon 132) and *IDH2* (codon 172) genes confirmed the IDH-wild type status of the tumor. Mutational analysis of the histone 3 genes (*H3F3A* and *HIST1H3B*) was negative, and *TERT C228T* and *C250T* promoter mutations were not identified. In addition, the tumor displayed strong nuclear p53 immunopositivity (Figure 3F), intense nuclear and cytoplasmic p16 expression, and retained nuclear expression of ATRX. Methylation-sensitive high-resolution melting confirmed the unmethylated status of the MGMT promoter. Tissue samples from the resection margins featured secondary structures of infiltrating glioma—the perivascular spreading and perineuronal satellitosis of tumor cells. The neoplasm was diagnosed as GBM, IDH-wildtype, WHO G4, according to the integrated diagnosis of the fifth edition of the WHO Classification of Tumors of the Central Nervous System 2021 [10].

Imaging prior to the oncological treatment uncovered GBM’s rapid early progression [2] (Figure 4A and Figure 5A). The patient received standard focal radiotherapy plus concomitant daily temozolomide followed by adjuvant temozolomide. Fractionated conformal radiotherapy was delivered using the volumetric modulated arc therapy (VMAT) technique to a total dose of 60 Gy in 30 daily fractions of 2 Gy each. Concomitant chemotherapy consisted of oral temozolomide at a daily dose of 75 mg/m^2^ given 7 days per week from the first to the last day of radiotherapy. After a 4-week break, the patient underwent only two cycles of adjuvant oral temozolomide for 5 days (first cycle 150 mg/m^2^ and second cycle 200 mg/m^2^) every 28 days. Although no sign of further progression of residual intracranial GBM was seen on the follow-up MRI 3 months after chemo-radiotherapy (Figure 4B and Figure 5B), the patient presented with the sudden onset of quadriparesis and paresthesia of the trunk and all extremities shortly afterward (five months after the diagnosis was established). MRI examination of the spine revealed intradural extramedullary spinal expansion at the C6 level with spinal cord compression and edema (Figure 6 and Figure 7). Laminectomy of C5–C7 and decompression of the spinal canal with biopsy were performed since radical resection was not possible. Histopathological examination displayed hypercellular glial neoplasm formed by plumb epithelioid cells with abundant pale eosinophilic cytoplasm and prominent nuclear pleiomorphism alternating with small cells with minimal cytoplasm and dense nuclear chromatin (Figure 8A). Tumor cells expressed GFAP and neuron-specific enolase (Figure 8D,F). The diagnosis of metastatic GBM infiltrating both the dura mater and the spinal cord was made based on the clinical presentation as an intradural extramedullary mass and microscopic similarities in the primary and metastatic tumor. Despite the hypofractionated course of palliative radiotherapy on the area of intraspinal infiltration (20 Gy in 5 fractions), the patient’s neurological condition deteriorated, and she was transferred to a palliative care institution after two weeks. Finally, she succumbed eight months after the surgery for the brain GBM and less than three months after the spinal GBM metastasis occurred.

To further apprehend the unique molecular features that might lead to extracranial GBM metastasis, next-generation sequencing (NGS) studies of both the primary and metastatic GBM were performed. DNA and RNA sequencing libraries were prepared from the primary GBM and metastatic tissue by employing the Illumina TruSight Tumor 170 assay and NextSeq 500 sequencer (Illumina Inc., San Diego, CA, USA). Data analysis was performed using the Illumina BaseSpace Sequence Hub, and the Variant Interpreter (Illumina’s Basespace tool) was used for the filtering and annotation of DNA variants. The custom variant filter was set up, including only variants with coding consequences and a GnomAD frequency value less than 0.01. Since the size of the TrueSight Tumor 170 panel was lower than 1 Mbp of exonic regions, the tumor mutational burden could not be accurately estimated. However, the number of single nucleotide variations (SNVs) observed in the metastatic sample was more than two times higher compared to the primary GBM. The detected gene variants of both the primary GBM and metastasis are listed in Table 1. Only one gene fusion of unknown significance, *EIF2B5-KIF5B*, was detected.

## 3. Discussion

In this work, we present a unique case of GBM with extracranial metastasis and its genetic background. In the described case, recurrent inactivating mutations in *TP53* (Gly245Ser), *PTEN* (Arg233Ter), and *RB1* (Trp563Ter) were observed in both samples, strongly supporting the metastatic origin of the spinal tumor. Alterations in these genes were indicative of the mesenchymal molecular subtype of GBM, typically associated with a poor prognosis. Furthermore, this gene alteration pattern was recently linked to glioblastoma with primitive neuronal component (GBM PNC) [11]. Similarly to the described case study, our case focally expressed synaptophysin (Figure 3D) and neuron-specific enolase (Figure 3E) and focally exhibited excessive proliferation activity Ki-67 (reaching up to 99%) (Figure 3C), and thus was reclassified as GBM PNC, despite featuring other morphological components such as epithelioid or giant cells. *MYC/NMYC* amplification, a common finding in GBM PNC [12], was not detected in either primary or metastatic samples. This rare GBM variant has a greater tendency toward leptomeningeal spread, which has been detected in up to 40% of cases [11]. Alterations in tumor suppressor genes *TP53*, *PTEN,* and *RB1* have been detected in 6 of 14 genetically evaluated metastatic GBM, including our case (43%). Among these, only one tumor was histologically diagnosed as GBM PNC, apart from the presented case [13,14,15,16,17]. On the other hand, this gene alteration pattern is not specific to GBM PNC since it was also revealed in gliosarcoma [13]. 

Apart from alteration in the p53 (apoptosis and senesce) and Rb pathways (cell-cycle progression), the described case harbored multiple gene alterations in the PI3K pathway, the main regulator of cell growth, survival, proliferation, invasion, and metastasis (Figure 9). A missense mutation in *MET* oncogene, the regulator of epithelial–mesenchymal transition, has been associated with elevated cell invasion, proliferation, and survival in atypical tissue environments. Furthermore, the *MET* gene was marked a signature gene of the mesenchymal molecular subtype [18,19]. 

To outline the possible molecular pathways of the metastatic spread of this particular GBM case, the gene mutation profiles of primary and metastatic tumors were compared. We suspected that the altered genes detected exclusively in the GBM metastasis are likely to be linked with the gain in the metastatic potential of GBM. During the disease progression, two inactivating mutations in *NF1* genes were acquired by the metastasizing GBM cells, which strongly supported and defined the mesenchymal molecular phenotype. The mesenchymal molecular subtype of GBM is known to be associated with a more aggressive, invasive, and recurrent course of the disease [20]. This suggestion is reinforced by the mesenchymal expression profile being frequently found in both GBM metastases as well as in circulating tumor cells [20,21]. Mutations in the *NF1* gene were described to be mutually exclusive of *BRAF* mutations in GBM [22], similar to what was detected in our case. *BRAF* mutation occurred in the primary tumor but was not detected in the metastasis, in which *NF1* gene mutations were found, suggesting that a subset of tumor cells lacking *BRAF* mutation gained the metastatic potential. Furthermore, the *BRAF* mutation detected in primary GBM leads to its loss of function. Loss of neurofibromin, a tumor suppressor protein encoded by *NF1*, results in the constitutive activation of the RAS cascade and its downstream effectors, including the MAPK, PI3K/AKT, and mTOR signaling pathways (Figure 9) [23,24]. Moreover, a mutation in the *MTOR* gene was also detected in the GBM metastasis. Apart from other effects, the mTOR signaling pathway induces epithelial–mesenchymal transition through the activation of transcription factors ZEB1/ZEB2, Twist, and Snail/Slug, the process in which tumor cells lose cell–cell adhesion molecules and obtain an augmented migratory phenotype [25]. The term glial–mesenchymal transition was coined, and it describes a similar process in GBM cells. Furthermore, its regulatory mechanism has been linked with transcription factors Snail, Slug, and Twist, with Snail being the master regulator [26]. The glial–mesenchymal transition process, along with cancer stem cells, has been associated with the expression of SOX2 as well as nestin and was detected in another metastatic GBM case [27], similar to the described case (Figure 3G and Figure 8G).

Furthermore, we found one likely pathogenic and one predicted-to-be-pathogenic variant of *NOTCH3* exclusively in the metastatic GBM sample. *NOTCH3* has a close relationship with metastatic spread in various cancers [28], including medulloblastoma, in which the activation of the NOTCH pathway was linked with distant metastatic disease and poor prognosis [29]. *NOTCH3* signaling results in the upregulation of matrix metalloproteinases that promote motility and invasion and is known to be one of the marker genes of the mesenchymal state and epithelial–mesenchymal transition [28,30] supporting the overall mesenchymal molecular phenotype and metastatic potential of the presented tumor. *NOTCH3* gene alterations have not been described in metastatic GBM yet; however, an altered *NOTCH1* gene was identified in lung GBM metastasis [13].

Although mutations in *ARID1A*, observed in our metastasis sample, are rare in GBM, with 0.7% in newly diagnosed cases, they are associated with an aggressive and mainly metastatic phenotype, as described in two other recent case reports [14,16]. The majority of recent data indicate that mutations in the *ARID1A* gene cause its loss of function, and its effect on tumorigenesis is context-dependent [31]. ARID1A, a subunit of the chromatin remodeling complex SWI/SNF, recruits mismatch repair protein, MSH2, to chromatin during DNA replication and thus promotes mismatch repair [32]. The metastatic localization of the mutated *ARID1A* gene in our case, along with the alteration in MSH6, thus signals the increased mutagenesis of metastatic cells, which supports the detected two-times-higher number of SNVs observed in the metastatic sample than in the primary GBM. Furthermore, ARID1A loss in established tumors accelerates tumor progression and metastasis, as was shown in the liver cancer model [32]. 

As mentioned above, the primary tumor was retrospectively sequenced by NGS and reclassified as GBM PNC based on the genetic alterations detected. In case the NGS testing was performed prior to the adjuvant oncological treatment, the oncological management would differ. MRI of the whole craniospinal axis would be performed, which is not routine in GBM patients. The detection of metastatic seeding would necessitate consideration of the whole craniospinal axis irradiation, which would not be considered otherwise.

## 4. Conclusions

Metastasis in GBM is a multistep process to which numerous genes collectively contribute and which is far from being fully understood yet. In this GBM PNC case, the alteration of *NF1*, *NOTCH3,* and *ARID1A* could explain, at least in part, the acquired invasiveness and metastatic potential, owing to the current knowledge. This work aims to highlight genetic alterations in one GBM PNC and its extracranial metastasis along with its likely contributions to tumor progression, highlighting the importance of tumor genetic testing and its impact on patient oncological management. 

## Figures and Tables

**Figure 1 diagnostics-13-00181-f001:**
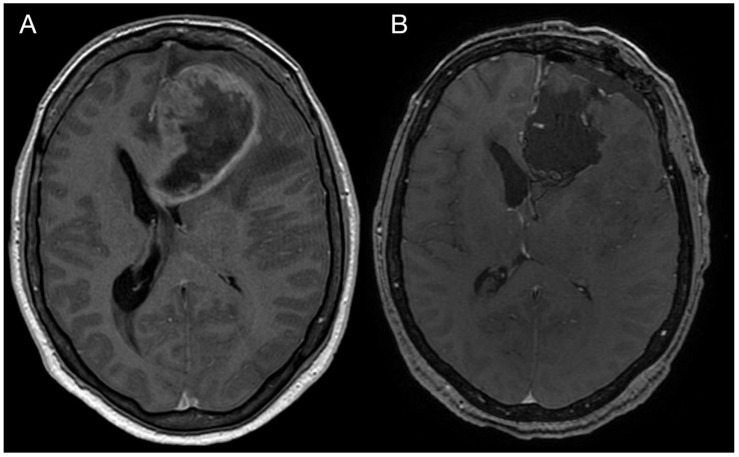
Imaging of the primary brain tumor. Preoperative axial brain MRI (postcontrast T1 weighted images) showing an intra-axial contrast-enhancing lesion in the left frontal lobe with perilesional edema and compression of the adjacent third and lateral cerebral ventricles (**A**). Postoperative MRI is showing total tumor resection. The resection cavity communicates with the frontal horn of the left lateral ventricle (**B**).

**Figure 2 diagnostics-13-00181-f002:**
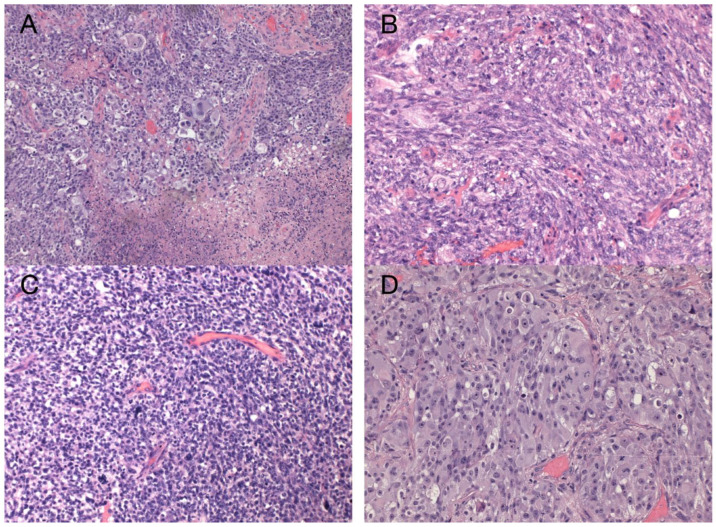
Histology of the primary GBM. Hypercellular high-grade diffuse glioma featuring significant cellular pleomorphism, including multinucleated giant cells, nuclear atypia, brisk mitotic activity, multiple microvascular proliferations, and necroses (**A**) (original magnification 100×). Regions with spindle cell morphology (**B**), small round blue cell morphology corresponding to regions of PNC (**C**), and epithelioid morphology (**D**) (original magnification 200×).

**Figure 3 diagnostics-13-00181-f003:**
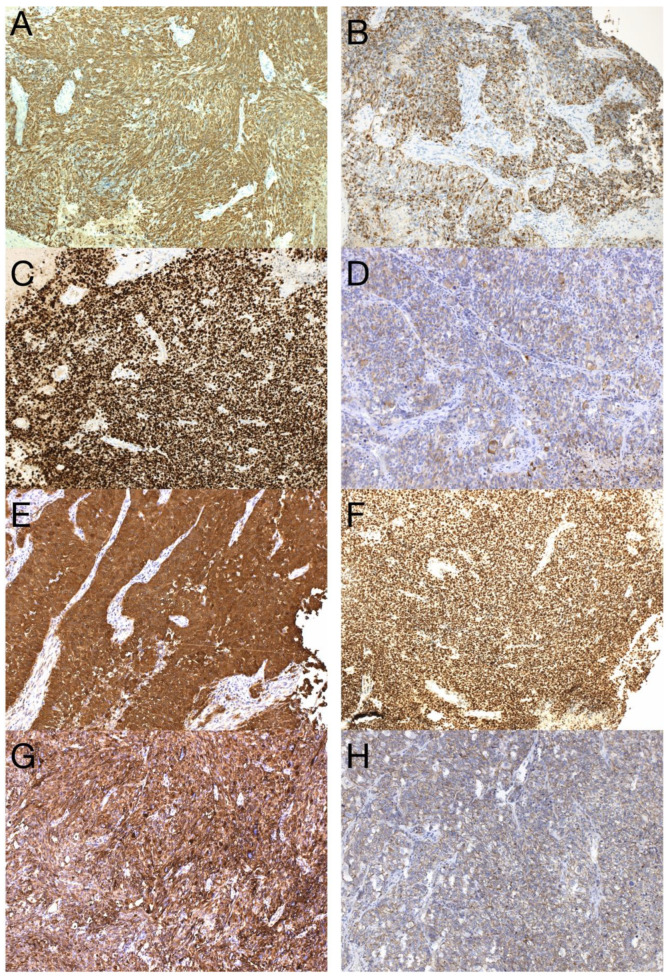
Immunohistochemical analysis of the primary tumor—GBM PNC, IDH wildtype. Immunohistochemical expression of GFAP in tumor cells (**A**), which was reduced in foci with PNC morphology (**B**). The proliferation index Ki-67 in hotspots morphologically corresponding to PNC reached up to 99% (**C**). In regions with PNC morphology, the expression of synaptophysin was detected immunohistochemically (**D**). The expression of neuron-specific enolase was observed throughout the tumor (**E**), as well as intense nuclear immunopositivity for p53 (**F**). GBM tumor cells displayed diffuse and strong expression of nestin (**G**). Beta-catenin expression was limited to the membrane, lacking nuclear expression (**H**) (original magnification 100×).

**Figure 4 diagnostics-13-00181-f004:**
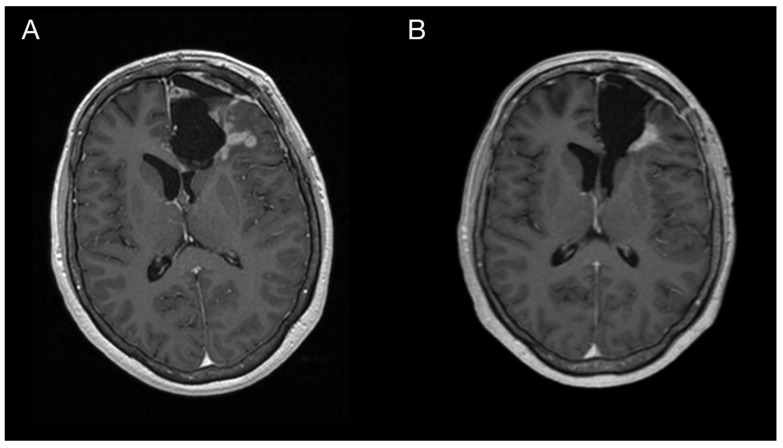
Imaging of the primary brain tumor prior to and after the oncological treatment. Axial brain MRI (postcontrast T1 weighted images) prior to oncological treatment displaying GBM’s rapid early progression in the left frontal lobe (**A**) and two months after the chemo-radiotherapy (**B**).

**Figure 5 diagnostics-13-00181-f005:**
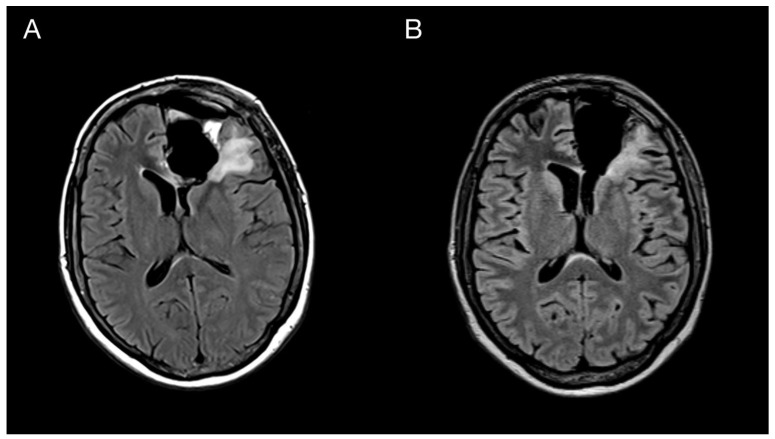
Imaging of the primary brain tumor prior to and after the oncological treatment. Axial brain MRI (fluid-attenuated inversion recovery images) prior to oncological treatment displaying GBM’s rapid early progression in the left frontal lobe (**A**) and two months after the chemo-radiotherapy (**B**).

**Figure 6 diagnostics-13-00181-f006:**
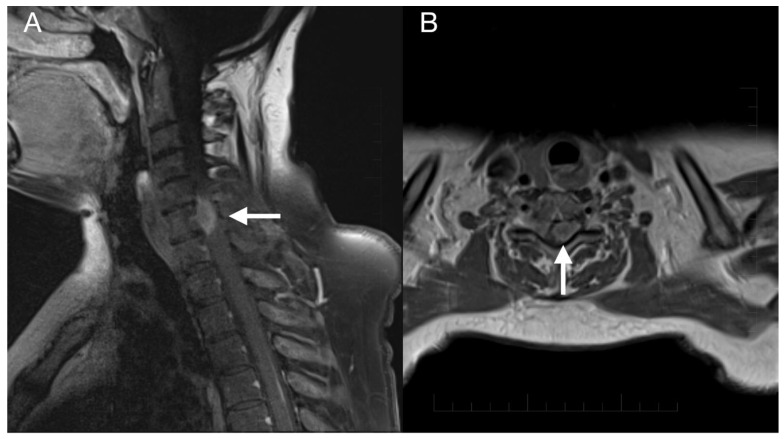
Imaging of the spinal metastasis. Five-month postoperative cervical spine MRI (postcontrast T1 weighted images) showing intradural expansion (arrow) at the C6 level intimately related to the spinal cord with spinal cord compression (**A**). Axial section showing intradural expansion (arrow) partially encircling the spinal cord and spreading into the neural foramen (**B**).

**Figure 7 diagnostics-13-00181-f007:**
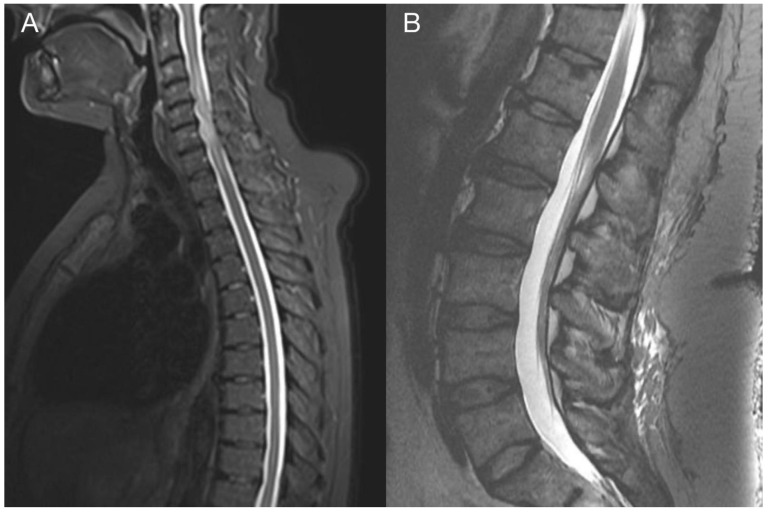
Imaging of the thoracic and lumbar spine. Preoperative MRI of the thoracic (**A**) and lumbar spine (**B**) did not show other pathological changes related to GBM metastasis.

**Figure 8 diagnostics-13-00181-f008:**
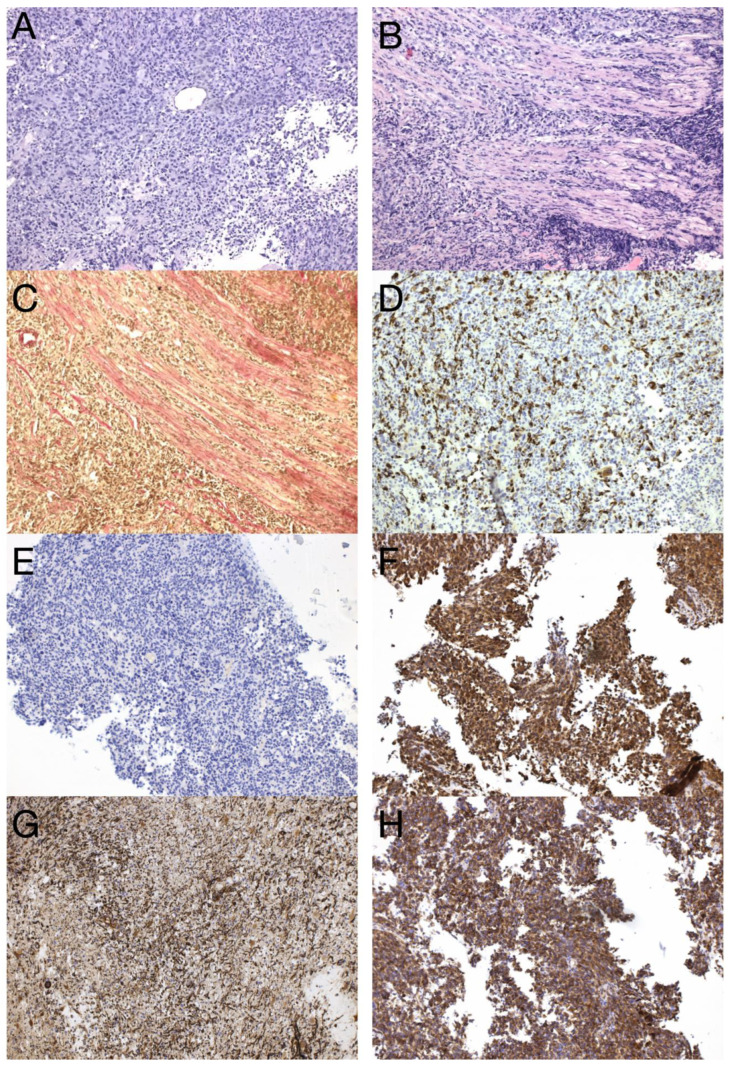
Histological and immunohistochemical analysis of GBM metastasis. Spinal metastasis—hypercellular high-grade diffuse glioma (**A**) infiltrating dura mater and spinal cord (**B**), verified by special staining—van Gieson, which stains collagen fibers red (**C**). Immunohistochemical expression of GFAP in tumor cells (**D**), synaptophysin (**E**), neuron-specific enolase (**F**), nestin (**G**), and beta-catenin displaying only membranous expression (**H**) (original magnification 100×).

**Figure 9 diagnostics-13-00181-f009:**
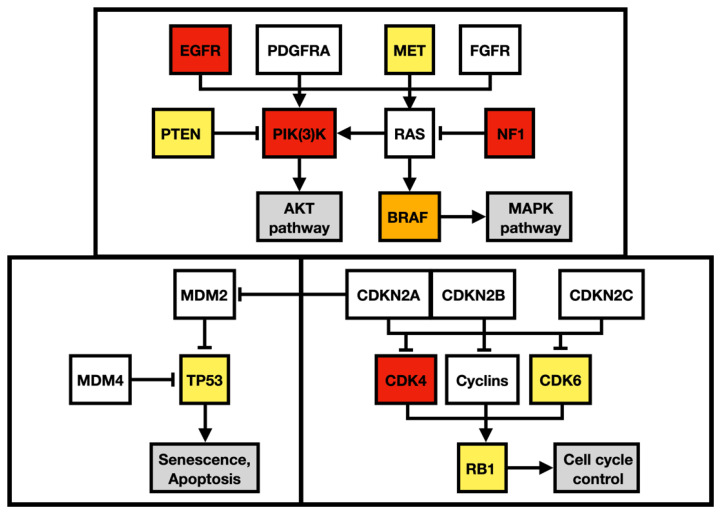
Regulatory pathway alterations in GBM. Altered genes in both primary and metastatic GBM are yellow, altered genes identified only in the primary GBM are orange, and altered genes detected only in the metastatic GBM are red. Adapted from Brennan et al. 2013 and Castellano et al. 2011.

**Table 1 diagnostics-13-00181-t001:** List of gene alterations detected by the NGS sequencing of the primary and metastatic GBM. Genetic alterations are separated based on their occurrence in the tumor samples. Yellow highlighted genes are genetic alterations shared by both primary and metastatic GBM. Altered genes identified only in the primary GBM are labeled orange. Altered genes detected only in the metastatic GBM are labeled red.

Gene Symbol	VID	cDOT	pDOT	Consequence	ACMG Classification
TP53	17:7577548:T	c.733G>A	p.Gly245Ser	missense_variant	Pathogenic
PTEN	10:89717672:T	c.697C>T	p.Arg233Ter	stop_gained	Pathogenic
RB1	13:48955573:A	c.1689G>A	p.Trp563Ter	stop_gained	Likely pathogenic
DHFR	5:79950728:79950727:CAGCGCCCC	c.420_428dup	-	5_prime_UTR_variant	Likely pathogenic
MET	7:116435786:G	c.3930A>G	p.Ile1310Met	missense_variant	Uncertain significance, some pathogenic evidence
INPP4B	4:143003277:A	c.2549C>T	p.Ser850Leu	missense_variant	Uncertain significance, likely pathogenic—minor evidence
CTNNB1	3:41266897:T	c.568C>T	p.Arg190Cys	missense_variant	Uncertain significance, likely pathogenic—minor evidence
CDK6	7:92247520:T	c.700G>A	p.Val234Met	missense_variant, splice_region_variant	Uncertain significance, likely pathogenic—minor evidence
BRAF	7:140508723:A	c.577G>T	p.Glu193Ter	stop_gained	Pathogenic
DNMT3A	2:25463224:C	c.2269A>G	p.Asn757Asp	missense_variant	Likely pathogenic
CSF1R	5:149440455:T	c.1939G>A	p.Val647Ile	missense_variant	Likely pathogenic
CARD11	7:2987212:T	c.217G>A	p.Ala73Thr	missense_variant	Uncertain significance, likely pathogenic—minor evidence
EP300	22:41572399:T	c.4928C>T	p.Ser1643Leu	missense_variant	Uncertain significance, some pathogenic evidence
NF1	17:29654741:A	c.5493G>A	p.Trp1831Ter	stop_gained	Pathogenic
CARD11	7:2984016:A	c.514G>T	p.Glu172Ter	stop_gained	Pathogenic
CREBBP	16:3779578:T	c.5470G>A	p.Ala1824Thr	missense_variant	Likely pathogenic
NOTCH3	19:15290914:T	c.3296G>A	p.Cys1099Tyr	missense_variant	Likely pathogenic
ARID1A	1:27094430:T	c.3138G>T	p.Arg1046Ser	missense_variant	Uncertain significance, likely pathogenic—minor evidence
MTOR	1:11259396:T	c.4172G>A	p.Arg1391Gln	missense_variant	Uncertain significance, likely pathogenic—minor evidence
EGFR	7:55233118:G	c.1868A>G	p.Asn623Ser	missense_variant	Uncertain significance, likely pathogenic—minor evidence
EGFR	7:55273060:T	c.3383C>T	p.Pro1128Leu	missense_variant	Uncertain significance, likely pathogenic—minor evidence
CHEK1	11:125507382:G	c.757A>G	p.Arg253Gly	missense_variant	Uncertain significance, likely pathogenic—minor evidence
CDK4	12:58144737:T	c.491T>A	p.Ile164Asn	missense_variant	Uncertain significance, likely pathogenic—minor evidence
SLX4	16:3651029:A	c.1114C>T	p.Arg372Trp	missense_variant	Uncertain significance, likely pathogenic—minor evidence
NF1	17:29557364:T	c.3077G>T	p.Arg1026Ile	missense_variant	Uncertain significance, likely pathogenic—minor evidence
RAD51C	17:56772457:A	c.311G>A	p.Cys104Tyr	missense_variant	Uncertain significance, likely pathogenic—minor evidence
NOTCH3	19:15285102:T	c.4513C>A	p.Pro1505Thr	missense_variant	Uncertain significance, likely pathogenic—minor evidence

## Data Availability

The data presented in this study are available on request from the corresponding author.

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
