# Peer review of "Spinal Metastasis in a Patient with Supratentorial Glioblastoma with Primitive Neuronal Component: A Case Report with Clinical and Molecular Evaluation"

_diagnostics, 2023, doi:10.3390/diagnostics13020181_

Round 1

Reviewer 1 Report

The article describes an interesting case of spinal metastasis from  IDH-wild type GBM. The authors report a very detailed clinical, neuroradiological, histological, and molecular description of the primary tumor and metastasis.  The discussion is also comprehensive and very interesting. I suggest some minor changes:

Line 114: The authors should better describe the chemo-radiotherapy protocol used, adding the dose of temozolomide during the concomitant and adjuvant phases, as well as the number of cycles of adjuvant chemotherapy administered before the spinal metastasis was found.

I suggest improving the quality of English. There are some errors in grammar and in the use of verb tenses (past and present used interchangeably).

Author Response

Dear reviewer

Thank you for the time and effort necessary to read our manuscript and prepare your comments and suggestions for how to improve our work. We appreciate all your constructive notes and comments and carefully considered all your recommendations to provide better and clearer text.
Herein, we explain how we revised the paper based on your comments and recommendations. We have addressed each as outlined below. All changes in the manuscript are highlighted.
We hope the revised manuscript will be accepted for publication in the recent form.

1) Line 114: The authors should better describe the chemo-radiotherapy protocol used, adding the dose of temozolomide during the concomitant and adjuvant phases, as well as the number of cycles of adjuvant chemotherapy administered before the spinal metastasis was found.

Hereby we would like to appreciate and thank you for such encouraging words. 
A detailed treatment description has been added:

”The patient received standard focal radiotherapy plus concomitant daily temozolomide followed by adjuvant temozolomide. Fractionated conformal radiotherapy was delivered using the volumetric modulated arc therapy (VMAT) technique to a total dose of 60 Gy in 30 daily fractions of 2 Gy each. Concomitant chemotherapy consisted of oral temozolomide at a daily dose of 75 mg/m² given 7 days per week from the first to the last day of radiotherapy. After a 4-week break, the patient underwent only two cycles of adjuvant oral temozolomide for 5 days every (first cycle 150 mg/m² and second cycle 200 mg/m²) 28 days.”

2) I suggest improving the quality of English. There are some errors in grammar and in the use of verb tenses (past and present used interchangeably).

We checked the text to improve the quality of the English.

Reviewer 2 Report

The manuscript by Hendrych and collaborators describes a rare case of glioblastoma metastasis to the spinal cord. As cases like these are extremely rare, the literature presents little data on these metastases and on their molecular profile. Thus, the present manuscript is quite interesting. I send some suggestions to improve the case description

1 - According to the new classification of brain tumors, all glioblastomas have IDH-wildtype. In this way, there is no need to say in the title that the glioblastoma found was IDH-wildtype.

2 - Authors should better describe in the case which findings (histological and molecular) led to the conclusion that the spinal cord tumor is a metastasis of the original tumor and not a new tumor

3 - On line 121 the authors say that the histopathological report identified that the tumor in the spinal cord was a metastatic GBM but does not describe the histopathological findings. Please include the histopathological characteristics of the tumor in the text.

4 - The authors need to better describe the results of table 1

Author Response

Dear reviewer

Thank you for the time and effort necessary to read our manuscript and prepare your comments and suggestions for how to improve our work. We appreciate all your constructive notes and comments and carefully considered all your recommendations to provide better and clearer text.
Herein, we explain how we revised the paper based on your comments and recommendations. We have addressed each as outlined below. All changes in the manuscript are highlighted.
We hope the revised manuscript will be accepted for publication in the recent form.

1) According to the new classification of brain tumors, all glioblastomas have IDH-wildtype. In this way, there is no need to say in the title that the glioblastoma found was IDH-wildtype.
We agree with your suggestion, and the redundant IDH-wildtype status has been removed from the title. 

2) Authors should better describe in the case which findings (histological and molecular) led to the conclusion that the spinal cord tumor is a metastasis of the original tumor and not a new tumor
Thank you for bringing this issue up, since distinguishing metastasis and metachronous tumor poses quite an obstacle in oncology.
In this case, the diagnosis of the metastatic tumor rather than the second primary tumor was concluded based on the clinical and histological presentation and was confirmed by NGS. The tumor mass was localized in intradural extramedullary spinal space and infiltrated the dura mater and the spinal cord. Since the exophytic growth in a diffuse glial neoplasm in both the brain and spinal cord is highly untypical, especially when most of the tumor mass grows outside of the spinal cord, the drop GBM metastasis is clinically more likely. The micromorphological similarity was detected in the primary and metastatic tumors, both presenting with epithelioid and small cell morphology (primitive neuronal component), which is quite rarely seen in GBM, supporting the likely metastatic origin of the spinal lesion. Our conclusion was confirmed by NGS testing. Detection of the same inactivating mutations in TP53 (Gly245Ser), PTEN (Arg233Ter), and RB1 (Trp563Ter) are very unlikely in different primary tumors. 
We have explained our reasoning in the text: 
Case report – “The diagnosis of metastatic GBM infiltrating both the dura mater and the spinal cord was made based on the clinical presentation as intradural extramedullary mass and microscopic similarities in the primary and metastatic tumors.”
Discussion – “In the described case, recurrent inactivating mutations in TP53 (Gly245Ser), PTEN (Arg233Ter), and RB1 (Trp563Ter) were observed in both samples, strongly supporting the metastatic origin of the spinal tumor. “

3) On line 121 the authors say that the histopathological report identified that the tumor in the spinal cord was a metastatic GBM but does not describe the histopathological findings. Please include the histopathological characteristics of the tumor in the text.
Thank you very much for spotting this detail which we have missed. 
A histological description has been added.:
“Histopathological examination displayed hypercellular glial neoplasm formed by plumb epithelioid cells with abundant pale eosinophilic cytoplasm and prominent nuclear pleiomorphism alternating with small cells with minimal cytoplasm and dense nuclear chromatin (Figure 8A). Tumor cells expressed GFAP and neuron-specific enolase (Figure 8D, F).”

4) The authors need to better describe the results of table 1
The description of Table 1 – “Gene alterations in the primary and metastatic GBM. Altered genes in both primary and metastatic GBM are yellow, altered genes identified only in the primary GBM are orange, and altered genes detected only in the metastatic GBM are red.” was replaced by “List of gene alterations detected by NGS sequencing of the primary and metastatic GBM. Genetic alterations are separated based on their occurrence in the tumor samples. Yellow highlighted genes are genetic alterations shared by both primary and metastatic GBM. Altered genes identified only in the primary GBM are labeled orange. Altered genes detected only in the metastatic GBM are labeled red.”